# Cannabidiol Antiproliferative Effect in Triple-Negative Breast Cancer MDA-MB-231 Cells Is Modulated by Its Physical State and by IGF-1

**DOI:** 10.3390/ijms23137145

**Published:** 2022-06-27

**Authors:** Alessia D’Aloia, Michela Ceriani, Renata Tisi, Simone Stucchi, Elena Sacco, Barbara Costa

**Affiliations:** 1Department of Biotechnology and Biosciences, University of Milano-Bicocca, Piazza della Scienza 2, 20126 Milan, Italy; michela.ceriani@unimib.it (M.C.); renata.tisi@unimib.it (R.T.); elena.sacco@unimib.it (E.S.); 2Milan Center for Neuroscience (NeuroMI), University of Milano-Bicocca, Piazza dell’Ateneo Nuovo 1, 20126 Milano, Italy; 3Department of Immunology and Microbiology, Lineberger Comprehensive Cancer Center, University of North Carolina, Chapel Hill, NC 27516, USA; simonstc@med.unc.edu; 4ISBE.IT, SYSBIO Centre of Systems Biology, Piazza della Scienza 2, 20126 Milan, Italy

**Keywords:** CBD, IGF-1 receptor, cisplatin, autophagy, action mechanism, bubbling cell death

## Abstract

Cannabidiol (CBD) is a non-psychoactive phytocannabinoid that has been discussed for its safety and efficacy in cancer treatments. For this reason, we have inquired into its use on triple-negative human breast cancer. Analyzing the biological effects of CBD on MDA-MB-231, we have demonstrated that both CBD dosage and serum concentrations in the culture medium influence its outcomes; furthermore, light scattering studies demonstrated that serum impacts the CBD aggregation state by acting as a surfactant agent. Pharmacological studies on CBD in combination with chemotherapeutic agents reveal that CBD possesses a protective action against the cytotoxic effect exerted by cisplatin on MDA-MB-231 grown in standard conditions. Furthermore, in a low serum condition (0.5%), starting from a threshold concentration (5 µM), CBD forms aggregates, exerts cytostatic antiproliferative outcomes, and promotes cell cycle arrest activating autophagy. At doses above the threshold, CBD exerts a highly cytotoxic effect inducing bubbling cell death. Finally, IGF-1 and EGF antagonize the antiproliferative effect of CBD protecting cells from harmful consequences of CBD aggregates. In conclusion, CBD effect is strongly associated with the physical state and concentration that reaches the treated cells, parameters not taken into account in most of the research papers.

## 1. Introduction

Among the various biological properties of phytocannabinoids, their ability to induce antiproliferative effects in different human cancer cells raises the scientific interest in their therapeutic potential in the field of oncology [1]. Concerning one of the most abundant non-psychoactive cannabinoids in *Cannabis sativa*, namely Cannabidiol (CBD), many in vitro and in vivo studies reported its effectiveness as an anti-cancer compound [2], although similar observations in humans are still missing. Despite this abundance of articles published over the last two decades following the pioneering study of Munson et al. [2,3] and the pilot screening of Ligresti et al. [4], a detailed analysis of the data highlights controversial and divergent findings [4,5], suggesting that its therapeutic potential as an anti-cancer drug in humans is still unclear. Focusing on in vitro studies, the effect of CBD on cancer cell viability ranges from no effect, to a modest reduction, and to significant cytotoxicity depending on concentrations, cancer cell lines, cell growth conditions, the performed assays, and the time of CBD exposure. CBD concentrations below three µM induced a decreased cell viability in human primary lung carcinoma cells [6] but failed in epithelial colon adenocarcinoma cells [7] or myeloma cells [8]; at higher concentrations, a general reduction in cell viability was observed, although the antiproliferative mechanisms vary from apoptosis to autophagy and cell-cycle arrest [9,10,11]. The assay conditions also affect CBD efficacy. Solinas et al. [12] reported that, in U87-MG and T98G glioma cells, the antiproliferative effects of CBD were similar under hypoxic and normoxic conditions, while Macpherson et al. [13] reported that Caco-2 colon cancer cells were more sensitive to CBD under conditions of physiological oxygen in the colon than those routinely used in cell culture experiments. In the same Caco-2 cells, a serum-dependent CBD effect on cell viability was observed, with a decrease in 5% serum and an increase in 1% serum [13]. Furthermore, a recent study on HT-29 human colorectal cancer cell line showed that CBD cytotoxic activity in cell culture media containing 10% serum is significantly reduced compared to a medium that contains only 0.5% serum, indicating an important influence of growth factors in CBD efficacy [14]. This wide pattern of results suggests that more mechanistic studies are required to clarify the therapeutic potential of CBD in cancer. To further characterize this issue, we analyzed the sensitivity of triple-negative human breast cancer cells to CBD, administered alone or in combination with antineoplastic drugs, in different growth conditions, revealing a role for IGF growth factor in CBD’s antiproliferative properties.

## 2. Results

### 2.1. Biological Effects of CBD on MDA-MB-231 Cells under Low and High Serum Growth Conditions

To study the effects of CBD on the MDA-MB-231 cell line, the ability of this phytocannabinoid to reduce cell proliferation at different time points (24, 48, and 72 h) was tested. Ligresti et al. [4] previously demonstrated that, among cannabinoids, CBD was the most potent in its antiproliferative activity and significantly affected the viability at 25 µM on MDA-MB-231 and MCF-7 cell lines.

At increasing CBD concentrations (10, 20, 30, and 50 µM), in a standard medium (10% FBS), cell growth was inhibited in the presence of 30 µM CBD (Figure 1A); furthermore, at this concentration we observed that some of the cells lose adherence. At 50 µM, the cell number dramatically decreases, already at 24 h.

Serum was previously reported to hinder the effect of CBD, but it is not clear if this effect could be attributable to the presence of components present in serum that compete with the cannabinoid for binding to receptors that mediate their activity [14,15]. Therefore, to elucidate CBD effects on the MDA-MB-231 cell line, we performed a kinetic growth assay at 24, 48, and 72 h since treatment, in a medium with low serum (0.5% FBS). The proliferative rate significantly decreases compared to DMSO when CBD is used at concentrations above 4 μM. In low serum, cells partially lose adherence at 5 μM, whereas this effect was reported only at 30 µM CBD in cells grown in 10% serum; at 10 and 20 μM, cells grown in low serum are mostly in the suspension, whereas this effect was observed only at 50 μM in the high-serum medium. At 10 μM, cells stop growing formally at 24 h from CBD treatment (Figure 1B).

This data suggests a threshold mechanism, rather than a dose-dependent curve, better describes the CBD effects, with a threshold that is reached between 3 and 5 μM.

In conclusion, cannabidiol reduces the proliferation of MDA-MB-231 cells in low serum at lower concentrations more than in high serum. This confirms that serum could contain some components that antagonize the effect of CBD itself.

To analyze if CBD has a cytotoxic or a cytostatic effect, supernatants from cells grown for 72 h in 0.5% FBS with different concentrations of CBD (10 or 20 µM) were collected, resuspended in a DMEM medium with 10% FBS without CBD, and plated in 60 mm tissue culture (TC) dishes. After five days, both 10 and 20 µM of CBD-pre-treated cells grew on new plates (Appendix A). Furthermore, to evaluate cell viability of adhering or detached cells after 72 h of CBD treatment in 0.5% serum, MTT assays were performed using two different methods: MTT solution was added directly to the cellular growth medium (Not Removed) or MTT solution was added to cells whose medium had been refreshed before (Removed). As shown in Figure 1C, cell viability decreases to 10% at a concentration of 5 µM only in the Removed samples, indicating that many cells are detached but not dead. At 10–20 µM, viability decreases in both samples, although the Removed one was always significantly lower compared to the corresponding Not Removed. These data demonstrate that CBD has cytostatic effects on MDA-MB-231 cells at 5 µM; at 10–20 µM, we observed both cytostatic and cytotoxic effects.

Subsequently, considering the observation of loss of adherence induction by CBD, we evaluated the ability of MDA-MB-231 cells to adhere to not-treated plates. Cells were pre-treated with 5 µM CBD for 72 h and then living cells were detached, collected, and plated in not-treated plates. Cells were cultured in a medium without CBD for 72 h. As it is shown in Figure 1D, cells can adhere to the new plate and, in particular, they are able to adhere more strongly than DMSO-treated cells, suggesting an effect on adherence properties of the treated cells, in agreement with a previous report, which showed that CBD pre-treatment induces an increase in ICAMs proteins [16].

To evaluate the effects of CBD on the MDA-MB-231 cellular transformation properties, we performed soft agar assays. Cells were previously treated with 5 µM CBD or DMSO for 72 h and then subjected to soft agar assays. Experiments have demonstrated that cells pre-treated with CBD are able to generate the same number of colonies as control cells (data not shown); analyzing the area, the perimeter, and the major and minor axis of colonies, it becomes clear that colonies are smaller, possibly due to treated cells having a delay in resuming growth (Figure 1E). Finally, to test the effects of CBD on cell migration, we performed a wound-healing assay on cells treated with 5 µM CBD or DMSO as described in the materials and methods section. Time-lapse data collected from three independent experiments at 10 h since wound creation shows that CBD does not influence migration and that there is no difference between control and sample in the percentage of wound coverage for any time analyzed (Figure 1F).

### 2.2. Antagonist Effect of CBD with Cisplatin in MDA-MB-231 Cell Line

Since triple-negative breast cancer is refractory to currently available targeted therapies and is generally treated with chemotherapy, we decided to evaluate the effect of CBD on MDA-MB-231 cells under the co-treatment with conventional chemotherapy agents. First, we evaluated the efficacy of cisplatin (DDP), doxorubicin (DOX), 5-fluorouracil (5 FU), and docetaxel on the viability of MDA-MB-231 cells (Appendix A). The analysis revealed a dose-dependent cytotoxic effect only for DDP and DOX, although used concentrations (micromolar) of DOX were significantly higher than the standardly used range (nanomolar). Since MDA-MB-231 resulted refractory to all chemotherapy agents tested except cisplatin, only this drug was used in co-treatment with CBD at sub-lethal concentrations.

MDA-MB-231 cells were treated for 72 h with different concentrations of CBD (0, 1, 10, 20, and 50 µM) and DDP (10 µM and 5 µM, corresponding to the value of IC50 and a sub-lethal dosage, respectively), alone or in combination, in a medium containing 10% FBS, and then subjected to the MTT assay (Figure 2). MTT results were analyzed using CompuSyn software, which performs quantitative analysis of the dose-effect relationship of multiple drugs. The analysis demonstrated that CBD exerts an antagonistic effect on DDP, as highlighted by the combination index value of the co-treatments greater than one (logCI > 0), except for the co-treatment with 50 µM CBD, where the CBD alone resulted highly toxic.

### 2.3. Effect of CBD on Autophagy, Apoptosis, and Induction of Bubbling Cell Death in the MDA-MB-231 Cell Line

Based on the previously obtained results, we wondered if the cells treated with CBD were arrested in a specific phase of the cell cycle. In fact, CBD induces G0-G1 phase cell cycle arrest in human gastric cancer SGC-7901 cells, by downregulating p21 protein expression and upregulating p53 protein expression [17].

To evaluate this aspect, we performed flow cytometry analyses following the treatment with CBD. First, we analyzed cells grown in 10% serum, since in the previous paragraph we demonstrated that in this condition the antagonism with the chemotherapeutic agent cisplatin increased. Thus, MDA-MB-231 cells were treated with CBD at concentrations of 10 and 20 µM in a medium with 10% serum for 48 h. As a control, we used the cells grown in a complete medium and those treated with DMSO. As can be seen in Figure 3A,B, the percentage of cells treated with CBD in the G1 phase is significantly higher compared to cells treated with the vehicle alone (DMSO). Consequently, MDA-MB-231 cells pause their cell cycle in the G1 phase at both tested concentrations of the cannabinoid.

Besides, we carried out FACS analyses of MDA-MB-231 cells grown in a medium with 0.5% serum and 5 µM CBD. The cells were treated for 48 h, whereas cells grown in the same medium without CBD and cells treated with the vehicle, DMSO, were used as a control. At 5 µM CBD concentration, a highly significant fraction of MDA-MB-231 cells arrests their cell cycle in the late S-G2 phase rather than in the G1 phase (Figure 3C,D).

Consistently with the growth kinetics results, it emerged that CBD has an antiproliferative action, causing a delay of the cell cycle in different phases, at different CBD/serum concentrations, suggesting that different mechanisms of action are expected for CBD antiproliferative effect and that one or more components of serum can counteract CBD action.

Considering the effect of CBD on growth and cell cycle arrest, we decided to carry out a molecular analysis on apoptosis and autophagy in the presence of CBD. Apoptosis and autophagy processes may coexist, antagonize or cooperate, balancing survival signaling versus death [18].

Shiravastava A. et al. [11] showed that beclin1, a core component of different complexes involved in the regulation of both autophagy and apoptosis [19], plays a central role in the induction of CBD-mediated apoptosis in MDA-MB-231 breast cancer cells grown in 10% FBS containing media. However, CBD activates mTORC1-independent and Unc51-like kinase (ULK1) dependent autophagy through ERK1/2 activation and AKT suppression [20].

Western blot analyses were performed on MDA-MB-231 cells grown for 48 h in medium with 10% or 0.5% serum in the presence of 5 µM (0.5% FBS) or 10 µM (10% FBS) CBD. In all the analyzed conditions, Caspase-3 protein was not cleaved. Therefore, in either condition, the cannabinoid was not able to activate the apoptotic process (Figure 4A,B).

To understand whether the treatment of the MDA-MB-231 cell line with the cannabinoid is able to trigger the autophagic mechanism, we evaluated the expression of LC3II on cells grown either in a medium with 10% FBS and treated with 10 µM CBD for 48 h or grown in a medium with 0.5% FBS and treated with 5 µM CBD. The expression of LC3II in the samples treated with 10 µM CBD (high serum) highlights the activation of the autophagic process but the expression ratio of the LC3II fragment/LC3I (normalized on control cells) in CBD-treated samples is not significantly affected when compared to the control (Figure 4A). Instead, autophagy is significatively activated in 5 µM CBD (low serum) treated cells compared to control cells, where the LC3II band is not visible (Figure 4B).

Cells treated with different concentrations of CBD (5, 10, and 20 µM) in DMEM containing 0.5% FBS show gas bubbles attached to the cells. These gas bubbles are particularly visible at all the analyzed concentrations of CBD, even if they are more abundant at 10 and 20 µM (Figure 4C). This phenomenon was firstly reported in 2015 by Chen SJ. et al. [21] and was named “bubbling cell death”. Bubbling death is an irreversible event and can be defined as ‘‘formation of a bubble from the nucleus per cell and release of this swelling bubble to the cell surface that ultimately causes cell death’’; its hallmark is the releasing of a single bubble per cell and hundreds of exosomes-like particles [22]. To better characterize this phenomenon we performed time-lapse analyses on Hoechst 33342-stained MDA-MB-231 cells with Operetta CLS™ (Figure 4D, video in Appendix A); images revealed an increase in bubbling that is dependent on CBD concentration and that is confirmed by quantitative analyses. At the concentration of 5 µM, there are few bubbles but autophagic vacuoles are visible inside the cells; at 10 µM bubbles occur in 80% of cells at 15 h from CBD addition, while at 20 µM the percentage of cells with gas bubbles reaches 85% already at 5 hours from CBD treatment. Likewise, there are many exosome-like particles visible in magnification.

### 2.4. Serum Interferes with the Threshold Effect of CBD on Cells

The above-described results highlight that CBD exerts a threshold-dependent effect on the cell viability of MDA-MB-231 cells and that the serum concentration in the culture medium changes the dose of CBD, at which this effect is observed. In fact, by lowering FBS concentration from 10 to 0.5% in the culture medium, the antiproliferative effect of CBD is observed at lower concentrations (Figure 1A,B). Therefore, we decided to investigate the mechanism, by which the serum exerts this modulatory effect on CBD action. Since CBD is a highly hydrophobic molecule with poor water solubility [23], we hypothesized that the serum could act as a surfactant by improving the solubility of CBD in the culture medium, preventing the formation of any aggregates/precipitates, and consequently affecting its bioavailability (namely the local concentration of CBD reaching the cells after supplementation to the culture medium) and biological activity. To verify this hypothesis, we analyzed by fluorometric assay the light scattering of culture media containing increasing concentrations of CBD (0–50 µM) in the presence of different serum concentrations, 0%, 0.5%, and 10% FBS, respectively (Figure 5A). The light scattering of samples at 540 nm (where the medium has low intrinsic emission) increased with the increase in the concentration of CBD, in a medium with low serum or without serum, suggesting that the formation of aggregates in these solutions are able to scatter the light. Notably, in a serum-free medium, where a lower solubility of CBD and greater precipitation are expected, the enhancement in the light scattering is observed at significantly lower concentrations. To confirm the hypothesis of the formation of CBD aggregates, we carried out cell viability assays on MDA-MB-231 cells treated for 72 h with different concentrations of CBD (5, 10, or 20 µM) in a medium containing 0.5% FBS, either as it is or subjected to filtration through membrane filters with 0.22 µm pore size, thereby capable of retaining aggregates with a size larger than the filter cut-off. From this analysis, it emerged that the antiproliferative effect exerted by the unfiltered CBD-containing medium is completely lost in the filtered medium (Figure 5B). Moreover, in cells treated with the filtered medium, a statistically significant enhancement of cell viability is observed compared to the control (cells treated with the vehicle), as measured by the MTT assay, similarly to that observed in the unfiltered medium with low concentrations of CBD, 1 and 3 µM (Appendix A). Actually, MTT assay detects the mitochondrial dehydrogenase activity of the analyzed cells, which is usually considered directly proportional to the number of metabolically active cells. Thus, the increase in absorbance recorded by the MTT assay in the cells treated with low concentrations of CBD could actually be attributable either to an increase in the number of viable cells or to an effect of CBD on cellular metabolism, an aspect that has not been described to date.

We then investigated whether the interfering effect of the serum on CBD action was also due to the presence of some components of the serum capable of either directly displacing the CBD from its cellular target/s or indirectly mitigating the antiproliferative effect exerted by CBD. To do this, we evaluated how increasing concentrations of the main growth factors present in serum (IGF-1, EGF, FGF-2, and TGFβ1) could complement the effect of CBD (5 µM) on cell viability of MDA-MB-231 cells in a medium containing 0.5% FBS (Figure 6A–D). IGF-2 growth factor, albeit contained in serum, was not included in the analysis since its specific receptor is not expressed in MDA-MB-231 cells [24,25]. As a positive control, we confirmed the ability of 10% serum to complement the effect of CBD (Figure 6E). Note in this case the pro-proliferative effect of 10% FBS compared to the control.

According to literature data, our analysis demonstrated that EGF is able to complement, albeit partially, the effect of 5 µM CBD (Figure 6B). In fact, CBD is known to inhibit EGF-induced activation of EGFR, proliferation, and chemotaxis in triple-negative breast cancer cells, including MDA-MB-231 [26].

The analysis also showed that, in addition to EGF, only IGF-1 was able to significantly complement, albeit only partially, the effect of CBD (Figure 6A). These data are consistent with data obtained on adhesion in the first paragraph, as IGF-1R can be found associated with E-cadherin and can stimulate adhesion interacting with the scaffold protein zonula occludens protein 1 (ZO-1) [27,28].

### 2.5. IGF-1 Antagonizes CBD in Triple-Negative Breast Cancer MDA-MB-231 Cells

To further investigate possible mechanisms of the specific effect observed by IGF-1 addition, a molecular docking simulation was performed on the only receptor for IGF-1 expressed in MDA-MB-231 cells, which is IGF-1R [24,25]. Both a structure for the inactive (PDB ID: 5u8r) and active (PDB ID: 6pyh) form were considered, since a massive conformational change is required for the receptor activation (Appendix A), which repositions the C-terminal α-helix (αCt in Figure 7A) from an intramolecular interaction (as in Figure 7A, left) to an intermolecular docking site in the dimer (as in Figure 7A, right), specifically on the L1 domain of the other monomer. This intermolecular docking is triggered and stabilized by the presence of IGF-1, which makes contact with the second monomer through its disordered central region, allowing the persistent activation of the receptor [29].

An initial docking survey was conducted on both active and inactive receptors, giving the main binding sites in similar positions, except for the different docking sites of the αCt element. The top poses were represented by the cavity surrounding the αCt (Figure 7A, left panel). For this reason, these docking sites were further investigated by flexible docking procedures, which confirmed IGF-1 binding to these sites (data not shown). For the active conformation structure, the IGF-1 was removed to evaluate the ability of CBD to occupy the same binding cavity as the natural ligand. These experiments revealed that CBD can make contact with the αCt in both conformations, in particular by inserting between the αCt of one monomer and the L1 domain of the other monomer in the active conformation (Figure 7A, top right), competing with the flexible linker in IGF-1 that is responsible for stabilization of the intermolecular interaction in the final conformation of the active receptor (Figure 7A, bottom right).

The overlay in the CBD binding site with the natural ligand-binding site (Appendix A) in the active form, but also the binding sites on the αCt in the inactive form, which represent indeed the first binding target of IGF-1, would tally with the competitive effect reported for IGF-1 and CBD in Figure 6A.

To confirm the ability of IGF-1 to antagonize the antiproliferative effect of CBD in MDA-MB-231 cells, growth kinetics under treatment with 5 µM CBD and IGF-1 50 µg/mL, alone or in combination, were performed in a medium containing 0.5% FBS. The number of viable cells were counted by the Trypan blue exclusion method every 24 h for 72 h (Figure 7B). The proliferation curves confirmed the ability of IGF-1 to partially rescue the cell viability of CBD-treated MDA-MB-231 cells. In particular, images acquired at a 24-h time point of the growth kinetics revealed that the autophagic vesicles induced by the 5 µM CBD treatment were completely abrogated by IGF-1 supplementation (Figure 7C). Finally, to explore the mechanistic effect (synergistic, antagonist, or additive) of IGF-1 on the action of CBD, the cell viability of MDA-MB-231 cells treated with different concentrations of CBD (1, 3, and 5 µM) and IGF-1 (10, 50, and 100 ng/mL), alone or in combination, were analyzed using CompuSyn software. The graphical representation of the obtained data (Figure 7D) indicated that IGF-1 at all the tested concentrations exerted an antagonist action on 5 µM CBD, having a value of log (combination index) greater than one. This result is also confirmed by morphological analysis at the inverted microscope (Appendix A), which shows that increasing concentrations of IGF-1 are able to partially counteract morphological changes induced by 5 µM CBD.

## 3. Discussion

Our work is contextualized in the discussion that is taking place in the scientific community on the safety and efficacy of the use of derivatives of *Cannabis sativa* in patients with cancer, in particular of cannabidiol, the most concentrated molecule in *Cannabis* devoid of psychotropic effects. Many research papers describe the antiproliferative effect exerted by cannabidiol on 2D and 3D preclinical cell models of cancer and also try to address the molecular mechanistic aspects of these effects [26,30,31,32]. In this context, we proposed to investigate the effect of CBD, alone and in co-treatment with conventional chemotherapeutic agents, on a cellular model of triple-negative breast cancer (MDA-MB-231 cell line), which is not responsive to the targeted therapies currently clinically available. Starting from literature data that highlight the role of CBD in interfering with cellular processes mediated by growth factor receptors, we decided to carry out assays not only in standard culture conditions, where the culture medium is supplemented with 10% serum (in our case FBS), but also in a low serum condition (0.5% FBS). This choice led us to understand that the effect exerted by CBD on cultured mammalian cells depends on the dose, which in turn impacts its state of aggregation and probably the concentration and/or the physical state that reaches the cells. In fact, the analyses highlighted very distinct behaviors in terms of cellular response associated with three different ranges of CBD concentrations with respect to a threshold concentration that is shifted based on the serum concentration in the culture medium, which acts as a surfactant agent (Figure 8).

Around the threshold dose, namely at 5 µM CBD in a medium with 0.5% FBS, a cytostatic antiproliferative effect is observed (Figure 1B,C,E), associated with a cell cycle arrest in G2 (Figure 3C,D), activation of autophagy (Figure 4B), and temporary loss of cell adhesion (Figure 1C), without influencing cell migration (Figure 1F), a typical behavior associated with the inhibition of signaling by growth factors such as IGF-1 [33,34,35,36]. Accordingly, the addition of IGF-1 partially recovers the viability of the cells treated with 5 µM CBD in a medium with 0.5% FBS (Figure 7). It should be noted that the cellular effects described above are attributable to the CBD in a state of aggregation, considering that the aggregates are completely lost following the filtration of the medium by membrane filters with a 0.22 µm size pore, as indicated in Figure 5B. Since it is not clear how CBD aggregates could specifically affect the IGF-1 receptor, we considered the hypothesis that the aggregates could facilitate the accumulation of CBD near the cell surface, leading to an increase in its efficacy in receptor binding.

In the absence of CBD aggregates, as in cells treated with a medium containing 5 µM CBD (or higher concentrations) with 0.5% FBS and filtered, a statistically significant promotion of cell viability or rather of metabolic activity, as detected with the MTT assay (Figure 5), is observed. The same is observed in cells treated with unfiltered medium containing low CBD concentrations (1 and 3 µM) with 0.5% FBS (Appendix A). The observed increase in the signal obtained with the MTT assay of the sample treated with low doses of CBD compared to the control, treated with the vehicle, is not associated with a higher number of cells, as emerged in the growth kinetics (Figure 1A), but could be associated with an increase in mitochondrial metabolism, which will need to be further investigated in future studies.

At doses above the threshold (doses > 5 µM in a medium with 0.5% FBS), CBD exerts a highly cytotoxic effect, which determines the phenomenon of bubbling death [21,22,37]. This is an irreversible process of death previously described only in UV-irradiated or cold-shocked cells; in these cells, apoptosis stops, and an enlarging nuclear gas bubble containing nitric oxide begins to form from the nucleus and is released to the cell surface causing cell death [22].

So, it is likely that at these high concentrations CBD forms colloidal aggregates, such as those observed in the standard medium at a concentration of 12.5 µM by Dynamic Light Scattering from Nelson and colleagues [38], capable of perturbing the permeability and integrity of the plasma membrane, thereby determining the bubbling. Finally, apoptosis is not activated in these cells as CBD switches the cell death mechanisms to bubbling cell death.

## 4. Materials and Methods

### 4.1. Drugs

Cannabidiol (CBD) was obtained by Cayman (Cayman Chemical, Ann Arbor, MI, USA). The pure cannabidiol was dissolved in dimethyl sulfoxide (DMSO) (Sigma-Aldrich, St. Louis, MO, USA) so that the final concentration of DMSO is equal to 0.05% in each experiment.

Cisplatin (DDP), 5-fluorouracil (5 FU), doxorubicin (Dox), and docetaxel were purchased from Sigma-Aldrich, Inc. (St. Louis, MO, USA). DDP was dissolved in phosphate-buffered saline (PBS1X, Euroclone, Pero, Italy) (stock solution 3.3 mM). 5FU and docetaxel were dissolved in dimethyl sulfoxide (DMSO) (Sigma-Aldrich, St. Louis, MO, USA) so that the final concentration of DMSO is equal to 0.05% in each experiment. Dox was dissolved in 4-(2-hydroxyethyl)piperazine-1-ethanesulfonic acid (HEPES, Sigma-Aldrich, St. Louis, MO, USA) (stock solution 1.7 mM).

### 4.2. Cell Culture

Triple-negative breast cancer cell line MDA-MB-231 was obtained from the American Type Culture Collection (ATCC) (LGC Standard, Teddington, UK). Cells were cultured in DMEM medium supplemented with heat-inactivated 10% foetal bovine serum (FBS), 2 mM L-glutamine, 100 U/mL penicillin, and 100 mg/mL streptomycin (Euroclone, Pero, Italy), at 37 °C in a humidified atmosphere of 5% CO_2_.

### 4.3. Cell Viability Trypan Blue Exclusion Assay

MDA-MB-231 cells were seeded in 24-well plates at a density of 1.4 × 10^5^ cells/mL. The day after, cells were treated with different concentrations of cannabidiol (CBD), and two different medium conditions were tested: (I) DMEM medium supplemented with 10% FBS; (II) DMEM medium supplemented with 0.5% FBS. After 24 h, cells were detached and counted as previously described [39]. In all experiments, unless otherwise specified, DMSO is referred to as control. The experiment was carried out in triplicate and repeated for three independent measurements. Data were analyzed using GraphPad v6.0 software (San Diego, CA, USA) employing two-way ANOVA followed by Dunnett’s test for group comparison. *p* < 0.05 was considered statistically significant.

### 4.4. Cell Viability MTT Assay

Cells were seeded in 96-well plates at a density of 2 × 10^5^ cells/mL. The day after, cells were treated or not with different concentrations of CBD (5, 10, 20 µM) in DMEM medium (100 µL) without phenol red and containing 0.5% FBS.72 h later, to test both adhesive properties and cell viability, MTT assay was performed in two ways: (#I) 10 µL of 3-(4,5-dimethylthiazol-2-yl)-2,5-diphenyltetrazolium bromide (MTT) stock solution (0.5 mg/mL; Sigma-Aldrich, St. Louis, MO, USA) was directly added to each well; (#II) cell culture medium was replaced with fresh one containing MTT. Cells were incubated at 37 °C for 4 h. After, formazan crystals were dissolved in acidic isopropanol. All the experiments were carried out in triplicate and repeated for three independent measurements.

### 4.5. Adhesive Properties Analysis

MDA-MB-231 cells were seeded in 60 mm plates at a density of 2 × 10^5^ cells/mL. The day after, cells were treated with different concentrations of CBD (5, 10, 20 µM) in DMEM medium supplemented with 0.5% FBS. Then 72 h later, detached cells (derived from 10 and 20 µM CBD treatment) were collected, centrifuged (800 rpm for 5 min), resuspended in DMEM medium with 10% FBS without CBD, and plated in 60 mm tissue culture (TC) dishes. Cells were observed with an inverted Olympus CKX41 microscope (Olympus Instruments, Tokyo, Japan). Instead, cells treated with 5 µM CBD were detached with trypsin (0.25%), collected, centrifuged (800 rpm for 5 min), resuspended in DMEM medium with 10% FBS without CBD, and plated at the density of 1.6 × 10^5^ cells/mL in 100 mm no-treated dishes. Cells were observed with an inverted Olympus CKX41 microscope (Olympus Instruments, Tokyo, Japan) and after 72 h of culture were detached with trypsin and counted. The experiments were carried out in triplicate and repeated for three independent measurements. Data were analyzed using GraphPad v6.0 software (San Diego, CA, USA) employing Student’s *t*-test. *p* < 0.05 was considered statistically significant.

### 4.6. Soft Agar Colony Formation Assay

To assess anchorage-independent growth, a soft agar colony formation assay was performed as previously described by Sacco et al. [40]. Briefly, MDA-MB-231 cells were seeded at the density of 2 × 10^5^ cells/mL in 60 mm dishes. The day after they were treated with 5 µM CBD or with DMSO (referred to as control) in a DMEM medium with 0.5% FBS for 72 h. Cells were then collected by trypsinization, counted by Trypan Blue exclusion method, and 50,000 live cells were plated in 60 mm tissue culture dishes containing 0.33% top agar and 0.5% bottom agar (Sigma-Aldrich, St. Louis, MO, USA). The medium was replaced every 3 days. After one month and a half, colonies were stained with MTT (Sigma-Aldrich, St. Louis, MO, USA), counted with an inverted Olympus CKX41 microscope, and characterized for their dimension with the use of ImageJ software (National Institute of Health, Bethesda, MD, USA). The experiment was done in triplicate and repeated for three independent measurements. Data were analyzed using GraphPad v6.0 software (San Diego, CA, USA) employing Student’s *t*-test. *p* < 0.05 was considered statistically significant.

### 4.7. Wound-Healing Assay

A wound-healing assay was performed as previously described by Pasquale et al. [41]. Briefly, 10^5^ cells were seeded in Cell Imaging 24-well Plates. The day after, cells were serum-starved (0.5% FBS) for 18h, and then a wound was made by scratching the monolayer cells; the medium was replaced with fresh 0.5% FBS medium containing 5 µM CBD or DMSO (referred to as control). Time-lapse imaging, performed by Operetta CLS™ (PerkinElmer, Inc, Waltham, MA, USA) equipped with a 10× air objective, was used to monitor wound coverage. The instrument was set up to capture images every hour for 24 h at 37 °C and 5% CO_2_. The experiment was carried out in triplicate and repeated for three independent measurements. Data were analyzed using GraphPad v6.0 software (San Diego, CA, USA) employing Student’s *t*-test. *p* < 0.05 was considered statistically significant.

### 4.8. Treatment with Chemotherapeutic Drugs

MDA-MB-231 cells were seeded in 96-well plates at a density of 2 × 10^5^ cells/mL. The day after, cells were treated with different concentrations of DDP (0, 5, 10, 20, 30,50 µM) or 5 FU (0, 10, 20, 40, 60, 100 µM) or Dox (0, 2, 5, 10, 20, 40, 60, 80, 100 µM) or docetaxel (0, 0.1, 1, 10, 100, 500, 1000, 1500, 2000) in DMEM with 10% FBS. 72 h later, cell viability was tested using an MTT assay (performed with method number I described above in the Cell Viability Assay (MTT assay section). The experiment was carried out in triplicate and repeated for three independent measurements. Data were analyzed using GraphPad v6.0 software (San Diego, CA, USA).

### 4.9. Co-Treatment with CBD and DDP

MDA-MB-231 cells were seeded in 96-well plates at a density of 2 × 10^5^ cells/mL. The day after, cells were treated with different concentrations of CBD (1, 5, 10, 20, 50 µM) and DDP (5, 10 µM), alone or in combination, in DMEM with 10% FBS. 72 h later, cell viability was tested using an MTT assay (performed with method number I described above in the Cell Viability Assay (MTT assay section). The experiment was carried out in triplicate and repeated for three independent measurements. Data were analyzed using GraphPad v6.0 software (San Diego, CA, USA) employing one-way ANOVA followed by Tukey’s test for group comparison. *p* < 0.05 was considered statistically significant.

### 4.10. Cell Cycle Analysis by Flow Cytometry

MDA-MB-231 cells were seeded at the density of 3 × 10^4^ cells/mL in 60 mm dishes. The day after, cells were treated with 5 µM CBD in DMEM containing 0.5% FBS or with two different concentrations of CBD (10, 20 µM) in DMEM containing 10% FBS. After 48 h, cells were fixed in 70% cold ethanol and incubated for 1h at 4 °C with PBS containing 50 µg/mL propidium iodide (Sigma-Aldrich, St. Louis, MO, USA) and 1 mg/mL RNase A (Sigma-Aldrich, St. Louis, MO, USA). Cell cycle perturbations were measured using a flow cytometer (CytoFLEX, Backman Coulter, Brea, CA, USA). At least 2 × 10^5^ cells were collected and evaluated for DNA content. Cell cycle distribution was analyzed using FlowJo 10 (BD Bioscience, NJ, USA).

### 4.11. Western Blot Analysis

MDA-MB-231 cells were seeded at the density of 2 × 10^5^ cells/mL in 60 mm dishes. The day after the medium was replaced or with DMEM 0.5% FBS, contained or not 5 µM CBD, or with DMEM 10% FBS, contained or not 10 µM CBD (DMSO is referred to as control). After 48 h from the treatment, protein extracts were prepared using lysis buffer containing 50 mM Tris–HCl, pH 7.4, 150 mM NaCl, 1% Triton X-100 *v/v*, 0.1 mM PMSF, 1 mM Na_3_VO_4_ (Sigma-Aldrich, St. Louis, MO, USA), supplemented with Complete ™ EDTA Free (Roche, Basel, Swiss) and 30 μg of total protein extracts were loaded and separated on 15% polyacrylamide gels. Western blotting was performed according to the standard protocol as previously described [41,42] using a nitrocellulose membrane. Blots were probed with anti-Caspase3 (1:1000), anti-LC3 (1:1000), and anti-GAPDH (1:3000) (Cell Signaling Technology, Danvers, MA, USA). Peroxidase-coupled donkey anti-rabbit IgG (Jackson Ltd., Philadelphia, PA, USA; dilution 1:5000) was used as a secondary antibody and immunoblots were developed using ECL Westar Nova 2.0 detection system (Cyanagen, Bologna, Italy). Data were analyzed using GraphPadv6.0 software (San Diego, CA, USA) employing the Wilcoxon test. *p* < 0.05 was considered statistically significant.

### 4.12. Bubbling Cell Death

MDA-MB-231 cells were seeded in 96-well plates at a density of 2 × 10^5^ cells/mL. The day after, cells were treated or not with different concentrations of CBD (5, 10, and 20 µM) in DMEM containing 0.5% FBS (DMSO is referred to as control). Then 24 h later, MDA-MB-231 cellular morphology was evaluated using an inverted Olympus CKX41 microscope (Olympus Instruments, Tokyo, Japan), equipped with a Digital C-Mount Camera TP 5100.

MDA-MB-231 cells were seeded in 96-well plates (uClear, lid black, Greiner Bio-One, Kremsmünster, Austria) at a density of 2 × 10^5^ cells/mL. The day after, cells were stained with Hoechst 33,342 (working concentration 1 µg/mL) (ThermoFisher, Waltham, MA, USA) for 15 min at 37 °C and 5% CO_2_, then treated as described above. Time-lapse imaging was performed using Operetta CLS™ (PerkinElmer, Inc, Waltham, MA, USA) equipped with a 40× immersion objective, as previously described [43]. The instrument was set up to capture images of each chosen field every 30 min, in brightfield and blue fluorescent channels, for 16 h at 37 °C and 5% CO_2_. For quantitative determination, the percentage of cells with gas bubbles was calculated. About 1500 cells derived from 10 different view fields of a plate for each sample were analyzed. The analysis was performed at the magnification described above. Total cell count (nuclei positive for Hoechst 33342) was obtained using the Harmony software while cells positive for gas bubbles were counted by eye. Data were analyzed using GraphPadv6.0 software (San Diego, CA, USA) employing ANOVA followed by Dunnett’s test for group comparison. *p* < 0.05 was considered statistically significant.

### 4.13. Light Scattering

CBD was diluted at different concentrations (0, 1, 2, 3, 4, 5, 10, 20, 30, 50 µM) in DMEM w/o Phenol Red (Gibco™-Thermo Fisher Scientific, Waltham, MA, USA) supplemented with 0, or 0.5 or 10% FBS. The light scattering of CBD-containing media was analyzed using a Cary Eclipse (Varian, Palo Alto, CA, USA). The light scattering measurements were carried out at 25 °C in a UV-cuvette by recording the spectrum in the 520–560 nm range at the excitation wavelength of 540 nm. The experiment was performed in two biological replicates. Spectra obtained in each medium, containing different serum concentrations, derived from a representative experiment were overlapped. The maximum light intensity registered at 540 nm was plotted as a function of CBD concentration.

### 4.14. Cell Viability MTT Assay in Presence of Filtered CBD

MDA-MB-231 cells were seeded in 96-well plates at a density of 2 × 10^5^ cells/mL. The day after, cells were treated or not with different concentrations of CBD (5, 10, 20 µM) as follows. CBD was added in DMEM medium (100 µL) without phenol red (containing 0.5% FBS) and then two distinct conditions were tested: (I) the cells were treated with this solution (same procedure utilized in the other experiments); (II) the solution was incubated at 37 °C for 20 min, then it was filtered and added to cells. The same procedure was performed for DMSO (referred to as control). Then 72 h later, cell viability was tested using an MTT assay (performed with method number I described above in the Cell Viability MTT Assay section). The experiment was carried out in triplicate and repeated for three independent measurements. Data were analyzed using GraphPad v6.0 software (San Diego, CA, USA) employing two-way ANOVA followed by Sidak’s test for group comparison and multiple Student’s *t*-tests. *p* < 0.05 was considered statistically significant.

### 4.15. Co-Treatment with CBD and Different Growth Factors

MDA-MB-231 cells were seeded in 96-well plates at a density of 2 × 10^5^ cells/mL. The day after, cells were treated with CBD (5µM) alone or in combination with IGF-1 (5, 20, 50 ng/mL), EGF (50 ng/mL), FGF-2 (5, 20, 50 ng/mL), TGFβ1 (1, 5, 10 ng/mL), or FBS (10%) in DMEM initially supplemented with 0.5% FBS. 48 h later, cell viability was tested using an MTT assay (performed with method number I described above in the Cell Viability Assay (MTT assay) section). The experiment was carried out in triplicate and repeated for three independent measurements. Data were analyzed using GraphPad v6.0 software (San Diego, CA, USA) employing one-way ANOVA followed by Dunnett’s test for group comparison or Student’s *t*-test. *p* < 0.05 was considered statistically significant.

### 4.16. Co-Treatment with CBD and IGF-1

MDA-MB-231 cells were seeded in 24-well plates at a density of 1.4 × 10^5^ cells/mL. The day after, cells were treated in the following two ways: (I) with CBD (5 µM) and IGF-1 (50 ng/mL) alone or in combination in a DMEM medium with 0.5% FBS and every 24 h, cells were detached and counted as described above (Trypan blue exclusion assay), (II) with different concentrations of CBD (1, 3, 5 µM) and IGF-1 (10 ng/mL, 50 ng/mL, 100ng/mL) alone or in combination and cell viability was tested using an MTT assay (performed with method number II described above in the Cell Viability Assay (MTT assay section) after 72 h.

### 4.17. Drug Combination Analysis

Viability data (MTT assay) derived from co-treatment with CBD and DDP or CBD and IGF-1 were converted to Fraction affected (Fa, range from 0 to 1, where Fa = 0 indicating 100% of cell viability while Fa = 1 indicating 0% of cell viability) and analyzed using the CompuSyn software (ComboSyn Inc., Paramus, NJ, USA) to calculate the combination index [log(CI)], being log(CI) < 0 an indication of synergism, log(CI) = 0 an indication of addictive effect and log(CI) > 0 an indication of antagonism.

### 4.18. Docking Analyses

To identify potential binding sites for CBD, an automated molecular-docking procedure was performed on the structure of both an inactive IGF-1 receptor (IGF-1R, PDB ID:5u8r), after removing the chain of the antibody included in the structure, and an active form of the same receptor (PDB ID: 6pyh), through the web-based SwissDock platform [44]. The docking was performed using the default parameters, with no region of interest defined (blind docking). The top score binding sites identified were further submitted to finer docking analysis in Maestro 10.1 (Schrödinger) (https://www.schrodinger.com/citations#Maestro accessed on 27 June 2022). All docking calculations were performed by the Glide software (Glide, version 6.7, Schrödinger, LLC, New York, NY, USA, 2015) according to the default procedure of the GLIDE docking eXtra Precision (XP) protocol. Finally, the best positions identified by the Glide XP protocol were used as the input for grid receptor definition in induced-fit docking (IFD) workflow with a flexible ligand option, allowing the protein to undergo sidechain movements during the docking (Schrödinger Suite 2015-2 Induced Fit Docking protocol; Glide version 6.7, Schrödinger, LLC, New York, NY, USA, 2015; Prime version 4.0, Schrödinger, LLC, New York, NY, USA, 2015). The IFD extended sampling protocol was employed. The OPLS 2005 force field was used for the minimization stage, with all parameters set as default except for the 7 Å limit of distance from each ligand pose for side chains optimization.

The structure of the inactive dimer of IGF-1R was obtained by HADDOCK 2.4 server (https://wenmr.science.uu.nl/haddock2.4/help accessed on 27 June 2022) by submitting the crystal structure (PDB ID: 5u8r) as a monomer and imposing the interface among the L2 of one monomer and the FnIII-1 domains of the opposing monomer, according to Xu et al. [29].

## 5. Conclusions

In summary, CBD in low doses does not exert any antiproliferative effect on MDA-MB-231 cells, but on the contrary, it could promote oxidative metabolism and other effects, possibly mediated by different growth factor receptors, yet to be elucidated. At intermediate doses, close to the threshold concentration, CBD induces survival mechanisms including cell cycle arrest and autophagy, and alteration of adhesive properties that could negatively interfere with the efficacy of conventional therapies and favor tumor invasiveness and progression. Accordingly, in MDA-MB-231, CBD exerts an antagonist action on cisplatin treatment (Figure 2). At high doses, on the other hand, CBD exerts a powerful cytotoxic action, by activating bubbling death [21].

Our results could explain the discordant data reported in the literature on the effect of CBD on healthy and cancer cells [4,23], since this effect is strongly associated with the physical state and concentration of CBD that actually reaches the treated cells, parameters not taken into account in most of the research papers. By changing these parameters, we were able to enlighten a specific effect of IGF-1 counteracting the CBD antiproliferative effect.

Future research will have to consider these aspects to define the mechanistic role of CBD, using cancer cellular models and possibly improving the use of nanoformulations to modify the CBD bioavailability [45,46,47] to better convey its desired concentration on the targeted cells.

## Figures and Tables

**Figure 1 ijms-23-07145-f001:**
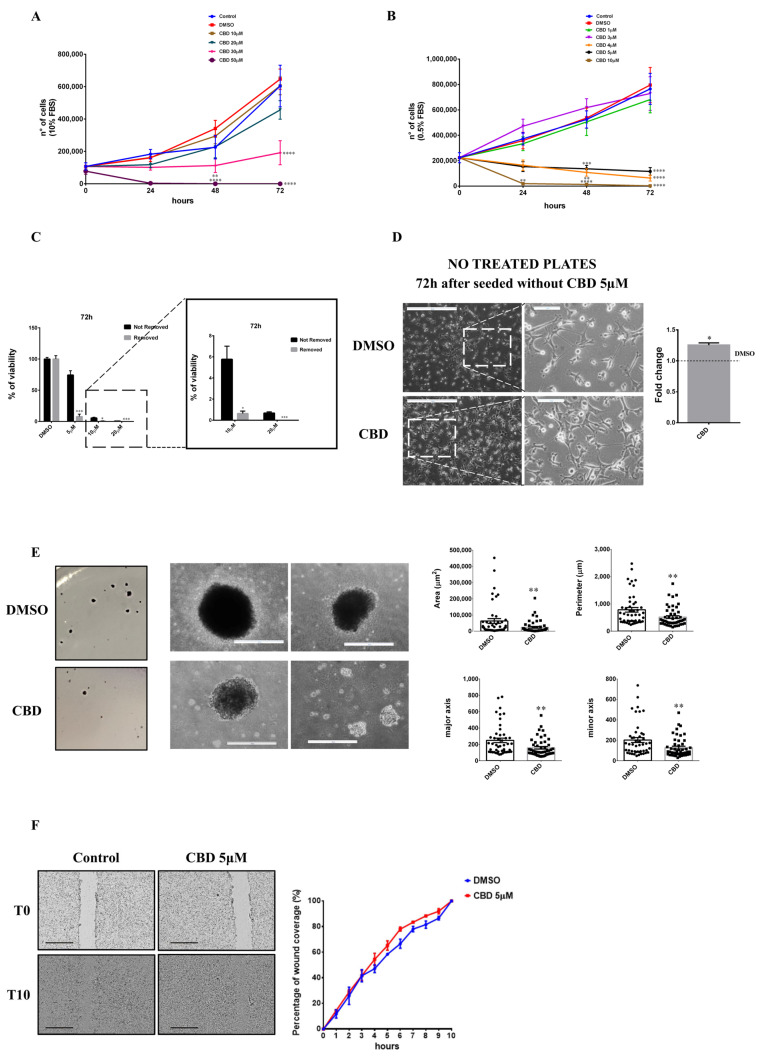
**Effects of CBD treatment on MDA-MB-231 cells.** MDA-MB-231 cells were plated on 24-well plates. After 24 h, cells were treated with different concentrations of CBD in DMEM medium supplemented with 10% FBS (**A**) or in DMEM medium supplemented with 0.5% FBS (**B**). The number of viable cells was determined at 0, 24, 48, and 72 h. (**C**) MDA-MB-231 cells were plated on 96-well plates. The day after, cells were treated with different concentrations of CBD in DMEM with 0.5% FBS. Cell viability was quantified after 72 h as described in Materials and Methods. Not Removed = MTT stock solution was added to each well without removing the culture medium. Removed = cell culture medium was removed and fresh medium containing MTT was added. (**D**) MDA-MB-231 cells were seeded and treated as described in Materials and Methods. After 72 h, cells were detached, collected, counted, and plated into not-treated dishes and optical images were captured. Representative images from at least three separate experiments are shown (left). Scale bar: 500 µm. Magnified views of cellular morphology are shown. Scale bar: 100 µm. Adhesive property is expressed as the ratio between the number of cells counted in the CBD pre-treated sample and the number of cells counted in the DMSO pre-treated sample (right). (**E**) Cell staining of colony formation in soft agar (left). Optical images of colonies were captured (right). Scale bar: 500 µm. Histograms relative to the dimension (area, perimeter, major and minor axis) of colonies are shown. (**F**) Representative wound healing assay images (left). Scale bar: 1 mm. On the right, the percentage of wound coverage is shown. Data are expressed as mean ± S.E.M. from three independent experiments. Differences among groups were analyzed using a one way analysis of variance (ANOVA) followed by Dunnett’s post hoc test (**A**,**B**) or *t*-tests (**C**–**F**) * *p* < 0.05, ** *p* < 0.01, *** *p* < 0.001, **** *p* < 0.0001.

**Figure 2 ijms-23-07145-f002:**
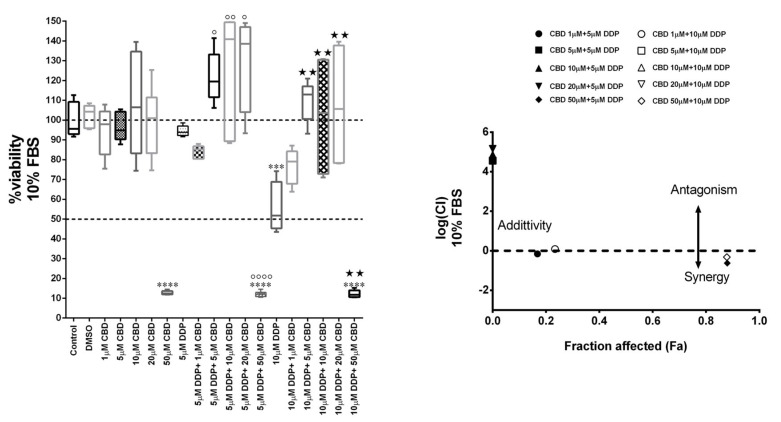
**CBD antagonizes the Cisplatin effect in MDA-MB-231 cells.** Cell viability in MDA-MB-231 cells co-treated with different concentrations of CBD and Cisplatin (DDP) in DMEM medium with 10% FBS. After 72 h from the treatment, the viability assay was measured by MTT assay (**left**). Data are expressed as mean ± S.E.M. from three independent experiments. Differences among groups were analyzed using a one-way analysis of variance (ANOVA) followed by Tukey’s post hoc test. *** *p* < 0.001, **** *p* < 0.0001 indicated significance compared to DMSO. ^∘^
*p* < 0.05, ^∘∘^
*p* < 0.01, ^∘∘∘∘^
*p* < 0.0001 indicated significance compared to 5 µM DDP. ^★★^
*p* < 0.01 indicated significance compared to 10 µM DDP. On the right, the logarithmic combination index plot is shown. Cell viability data derived from the MTT assay described above was converted to Fraction affected (Fa) values and analyzed by CompuSyn software to obtain a combination index plot. log(CI) < 0, synergism; log(CI) = 0, addictive effect; log(CI) > 0, antagonism. Hallow box and triangles (CBD 5 µM + 10 µM DPP, CBD 10 µM + 10 µM DPP, CBD 20 µM + 10 µM DPP) are covered by black box and triangles (CBD 5 µM + 5 µM DPP, CBD 10 µM + 5 µM DPP, CBD 20 µM + 5 µM DPP).

**Figure 3 ijms-23-07145-f003:**
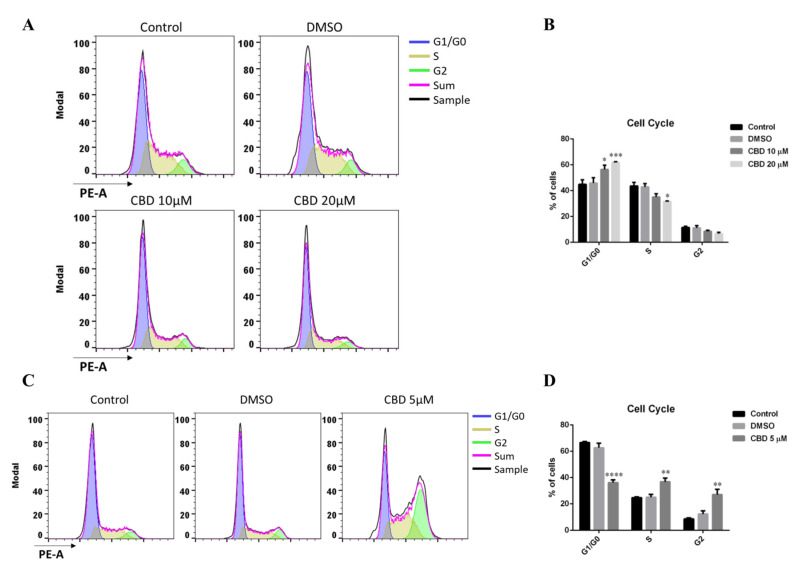
**CBD induces cell cycle arrest.** (**A**,**B**) MDA-MB-231 cells were cultured using DMEM supplemented with 10% FBS. 24 h later, cells were treated with 10 or 20 µM of CBD. After 48 h, cells were collected and used for flow cytometry assay. (**A**) Representative histograms of gated cells in G1/G0 (blue), S (gold), and G2 (green) phases. Plots and analyses were made using FlowJo 10 software. (**B**) Quantitative analyses of distributions or proportions of the cells in each phase. (**C**,**D**) MDA-MB-231 cells were cultured using DMEM supplemented with 0.5% FBS. Then 24 h later, cells were treated with 5 µM of CBD. After 48 h, cells were treated as described in (**A**,**B**). (**C**) Representative histograms of gated cells in G1/G0 (blue), S (gold), and G2 (green) phases. Plots and analyses were made using FlowJo 10 software. (**D**) Quantitative analyses of distributions or proportions of the cells in each phase. Each bar represents the mean ± S.E.M. of data from at least three independent experiments. Differences among groups were analyzed performing a two-way ANOVA followed by Tukey’s multiple comparison test. * *p* < 0.05, ** *p* < 0.01, *** *p* < 0.001, **** *p* < 0.0001 indicates significance compared to DMSO.

**Figure 4 ijms-23-07145-f004:**
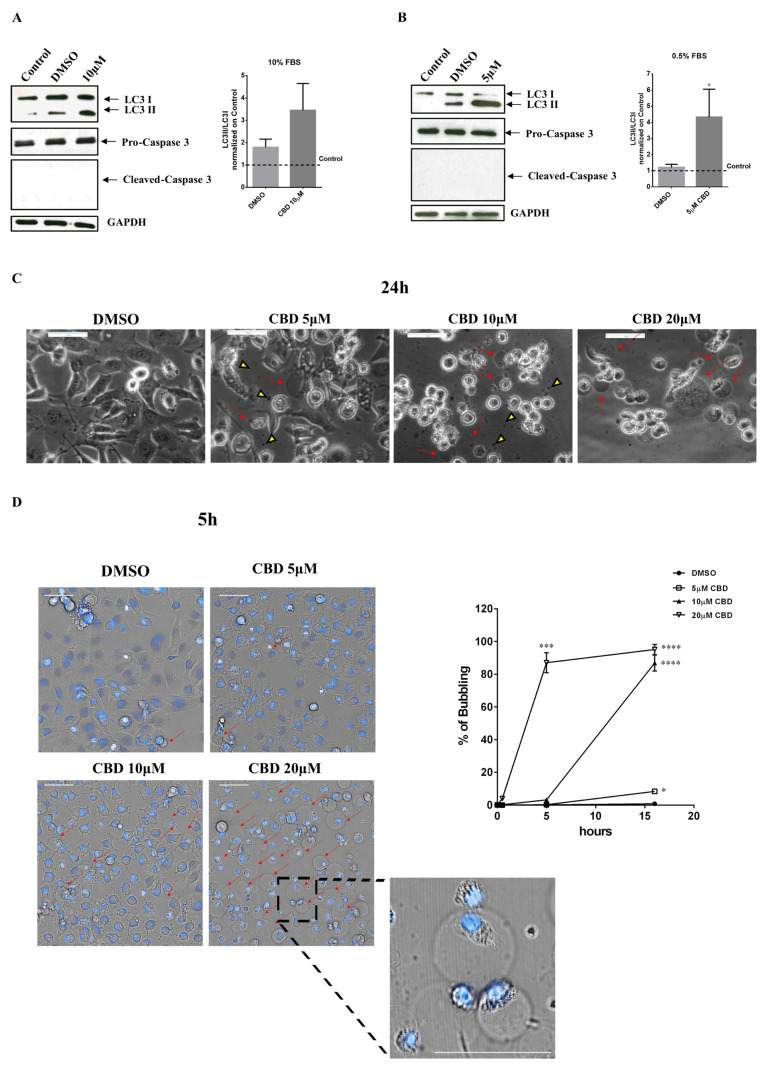
**CBD induces autophagy and bubbling cell death in low serum conditions.** MDA-MB-231 cells were seeded, and the day after, the medium was replaced either: (**A**) with DMEM 10% FBS, containing or not 10 µM CBD; or (**B**) with DMEM 0.5% FBS, containing or not 5 µM CBD (DMSO is referred to as control). Cell lysates were separated on SDS-PAGE and blotted to nitrocellulose membrane; blots were probed with anti-LC3, anti-caspase3 or anti-GAPDH antibodies. GAPDH was used as a loading control. Data are expressed as mean ± S.E.M. from three independent experiments. Differences among groups were analyzed using the Wilcoxon test. * *p* < 0.05. (**C**) MDA-MB-231 cells were seeded and treated as described in Materials and Methods. 24 h after the treatment, optical images were captured. Scale bar: 50 µm. Red arrows indicate gas bubbles while yellow triangles indicate exosomes. (**D**) MDA-MB-231 cells were seeded in uClear black 96-well plates. After 24 h, cells were stained with Hoechst (blue), treated with the indicated concentrations of CBD in DMEM 0.5% FBS, and time-lapse imaging was performed using Operetta CLS™ equipped with a 40× immersion objective. On the left, cellular morphology of MDA-MB-231 after 5 hours from the treatment. Scale bar: 50 µm. Red arrows indicate gas bubbles. A magnified view of gas bubbles is shown. On the right, the percentage of cells positive for gas bubbles at different time points is shown. Data are expressed as mean ± S.E.M. from three independent experiments. Differences among groups were analyzed using a one-way analysis of variance (ANOVA) followed by Dunnett’s post hoc test. * *p* < 0.05, *** *p* < 0.001, **** *p* < 0.0001.

**Figure 5 ijms-23-07145-f005:**
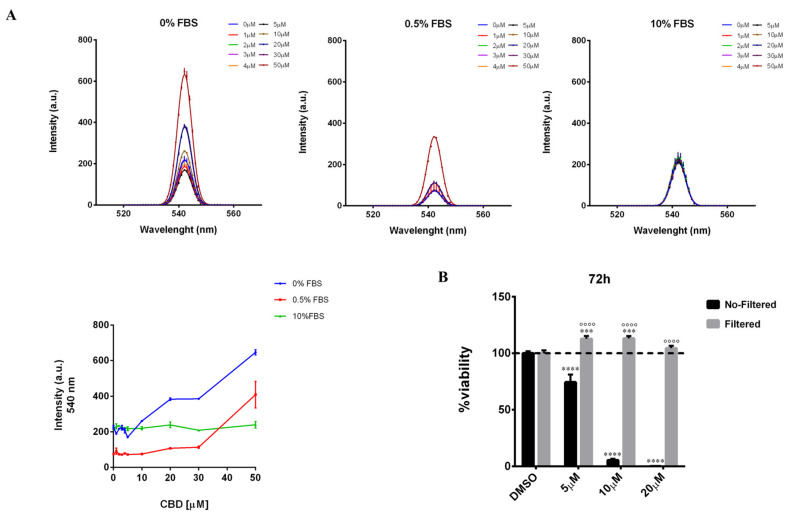
**Serum exerts a surfactant action on CBD-aggregates.** (**A**) Light scattering spectra of increasing concentrations of CBD in DMEM medium (without phenol red) supplemented with different concentrations of serum (0% FBS, 0.5% FBS, 10% FBS). On the bottom left, a graphical representation of the light scattering of the samples at 540 nm. (**B**) MDA-MB-231 cells were plated on 96-well plates. The day after, cells were treated as described in Materials and Methods. Cell viability was quantified after 72 h of the treatment. No-Filtered = CBD was added in DMEM medium without phenol red (containing 0.5% FBS) and then cells were treated with this solution. Filtered = CBD was added in DMEM medium without phenol red (containing 0.5% FBS); after that, the solution was incubated at 37 °C for 20 min, then it was filtered and added to cells. Data are expressed as mean ± S.E.M. from three independent experiments. Differences among groups were analyzed using two-way ANOVA followed by Sidak’s test for group comparison and multiple *t*-tests. *** *p* < 0.001, **** *p* < 0.0001 indicated significance compared to DMSO. ^∘∘∘∘^
*p* < 0.0001 indicated significance compared to the No-Filtered sample at the same concentration.

**Figure 6 ijms-23-07145-f006:**
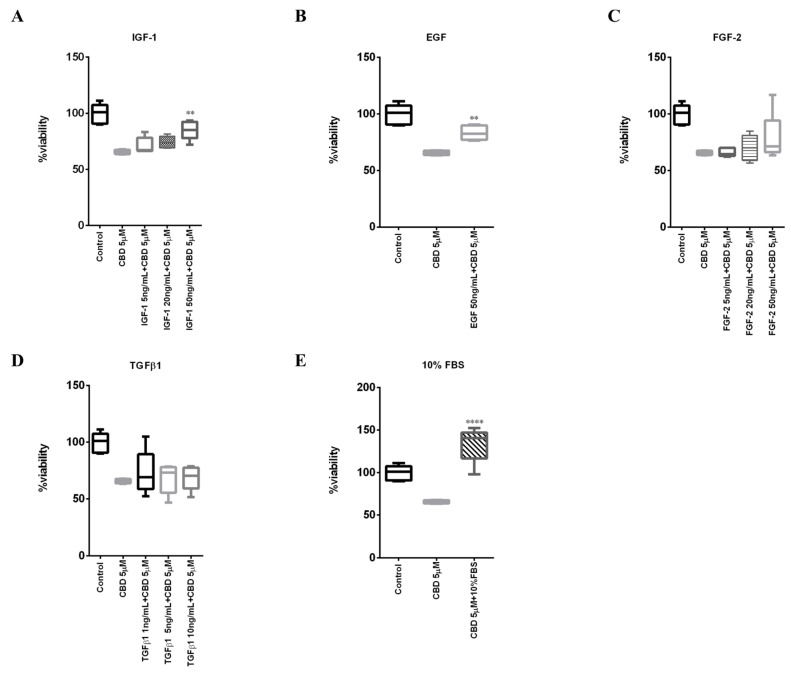
**IGF-1 antagonizes CBD’s effect on cell viability.** MDA-MB-231 cells were plated on 96-well plates. The day after, cells were treated with 5 µM CBD in combination with different concentration of (**A**) IGF-1, (**B**) EGF, (**C**) FGF-2, (**D**) TGFβ1 or with (**E**) 10% FBS in DMEM with 0.5% FBS. Cell viability was quantified after 48h as described in Materials and Methods. Data are represented with box and whisker plots derived from three independent experiments. Differences among groups were analyzed using one-way ANOVA followed by Dunnett’s post hoc test. ** *p* < 0.01, **** *p* < 0.0001.

**Figure 7 ijms-23-07145-f007:**
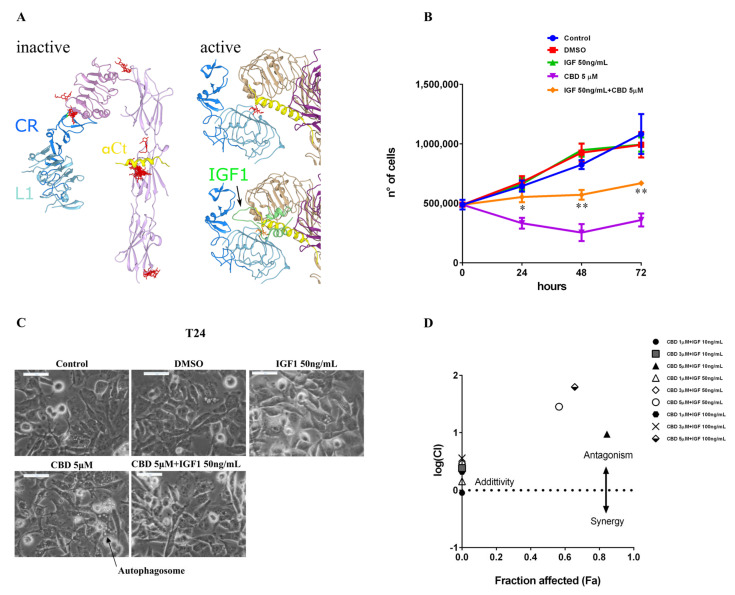
**Docking experiments suggest that CBD can bind to overlapping sites with IGF-1.** (**A**) Domains involved in the intermolecular interaction upon receptor activation are differently colored for identification: L1 domain is light blue, CR connector is blue, αCt helix is yellow. CBD molecules are represented as red sticks. Left, the best poses of the top 20 clusters were reported for the Swissdock docking survey with CBD as a ligand and an inactive IGF-1R structure (PDB ID: 5u8r, minus the antibody chains) as a receptor. Top right, best pose from the flexible docking experiment on an active structure of IGF-1R (PDB ID: 6pyh, minus the natural ligand, IGF-1). The two different monomers are colored in purple (the αCt belongs to this monomer) and tan (the visible L1 and CR belong to this monomer). Bottom right, the IGF-1 (green) position in the active receptor structure (6pyh) is shown, overlapping the CBD docking site. (**B**) MDA-MB-231 cells were plated on 24-well plates. After 24 h, cells were treated with CBD (5 µM) alone or in combination with IGF-1 in a DMEM medium supplemented with 0.5% FBS. The number of viable cells was determined via Trypan blue staining at 0, 24, 48, and 72 h. Data are expressed as mean ± S.E.M. from three independent experiments. Differences among groups were analyzed using multiple *t*-tests. * *p* < 0.05, ** *p* < 0.01 indicates significance between IGF-1 50 ng/mL + CBD 5 µM and CBD 5 µM. (**C**) Cellular morphology of MDA-MB-231 cells treated as described above and after 24 h optical images were captured. Scale bar: 50 µm. (**D**) MDA-MB-231 cells were seeded in 96-well plates. The day after, cells were treated with different concentrations of CBD (1, 3, 5 µM) and IGF-1 (10 ng/mL, 50 ng/mL, 100 ng/mL), alone or in combination. Then 72 h later, cell viability was tested using MTT assay as described in Materials and Methods. Cell viability data were converted to Fraction affected (Fa) values and analyzed by CompuSyn software to obtain a combination index plot. log(CI) < 0, synergism; log(CI) = 0, addictive effect; log(CI) > 0, antagonism. Black hexagon (CBD 1 µM + IGF 100 ng/mL) is partially covered by gray box (CBD 3 µM + IGF 10 ng/mL).

**Figure 8 ijms-23-07145-f008:**
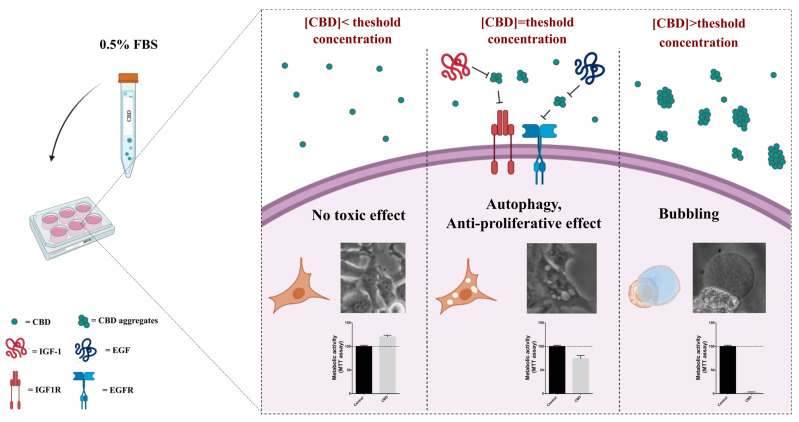
Schematic representation of biological effects exerted by CBD on cells.

## Data Availability

Not applicable.

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
