# Peer review of "Cannabidiol Antiproliferative Effect in Triple-Negative Breast Cancer MDA-MB-231 Cells Is Modulated by Its Physical State and by IGF-1"

_ijms, 2022, doi:10.3390/ijms23137145_

Round 1

Reviewer 1 Report

This is an excellent paper! I really enjoy reading it. The authors have investigated the effects of CBD on a triple negative breast cancer cell line, not responsive to the targeted therapies currently clinically available. They’ve examined aspects of CBD anticancer effects by conducting pertinent experiments that answer to to-the-point questions previously imposed in the contemporary literature, and interesting new ones. The results and the conclusions supported by them are crystal clearly presented, answering to the imposed questions and generating new interesting hypotheses. Therefore, I strongly recommend the publication of the manuscript in IJMS with only the following minor recommendations:

p.4-5, caption of Figure 1: I think that technical details like “Optical images of colonies were captured with an inverted Olympus CKX41 microscope” (repeated twice in this caption) or/and “(C) MDA-MB-231 cells were plated on 96-well plates. The day after, …” that are also given in the “Materials and Methods” section, could be avoided in the already overloaded captions of Figures. In general, I suggest to suppress the unnecessary details, which are also given in “Materials and Methods”, from all figures’ captions (e.g. in Figure 3 “Differences among groups were analyzed using GraphPad Prism software performing a two-way ANOVA followed by Tukey’s multiple comparison test” that is repeated twice).

p. 14, caption of Figure 7, lines 6-7 “The residues involved in contact with CBD molecules are represented as green sticks and labeled in black.” This cannot be seen in the picture, better suppress this sentence.

4. Materials and Methods

p.18 “4.6. Soft agar colony formation assay”, 8th line of this section: Move comma “After one month, and a half colonies …” to “After one month and a half, colonies …”

p.20, 2nd ln: Better replace “by eyes” with “by eye”

p.20 “4.13 Light scattering”:  Correct “CDB” with “CBD” in the 1st and 3rd line of this subsection

p.20 “4.14. Cell Viability MTT Assay in presence of filtered CBD”, 2nd-6th lns: The repetition is rather confusing. I suggest: “The day after, cells were treated or not with different concentrations of CBD (5,10,20 μM) as follows. CBD was added in DMEM medium (100μl) without phenol red (containing 0.5% FBS) and then two distinct conditions were tested: (I) the cells were treated with this solution (same procedure utilized in the other experiments); (II) the solution was incubated at 37°C for 20 min, then it was filtered and added to cells.”

p.20 “4.15. Co-treatment with CBD and different growth factors”, 4th ln of this subsection: Maybe the addition of somewhat like “… in DMEM initially supplemented with 0.5% FBS” will remove the confusion made by a first glance in the sentence “… FBS (10%) in DMEM supplemented with 0.5% FBS”

p. 21 “4.18 Docking analyses”, 1st line of this subsection: Better replace “ … binding sites of CBD” with “ … binding sites for CBD” as these are binding sites of IGF-1R

p.20 “4.16. Co-treatment with CBD and IGF-1”, 3rd line of this subsection: Better replace “At each 24h” with “Every 24h”

p.21 1st line: “MDA-MB-231 cells were seeded in 96-well plates at a density of 2 x 105 cells/mL.” Is this repetition unavoidable? For example, the 4.16 subsection could start with “MDA-MB-231 cells were seeded in two 96-well plates at a density of 2 x 105 cells/mL. The day after, cells were treated in the following two ways: I) with CBD … II) with different concentrations of CBD …”

p 21, 3rd – 5th lines: Better replace “ … alone or in combination. 72h later, cell viability was tested using ...” with “ … alone or in combination and cell viability was tested using an MTT ... after 72h”

p.22 last sentence “use of nanoformulations to modify  the CBD bioavailability”. What about cyclodextrins? See for example Pharmaceutics 2022, 14, 706. https://doi.org/10.3390/pharmaceutics14040706 ;)

Since the sentences ”Cells were observed with an inverted Olympus CKX41 …” and “Data were analyzed using GraphPad …” are constantly repeated in the subsections, maybe it would be a good idea to add a final subsection entitled “Cells observation and Data Analysis” and remove the repeated text from all the subsections that these sentences are repeated. Of course, the new subsection should contain “two-way ANOVA followed by Dunnett’s test for group comparison was employed for Cell viability Trypan blue exclusion assay (4.3); Student’s t-test was employed for Adhesive properties analysis (4.5); …. etc". On the other hand, the repetition, although boring, sometimes offers clarity (especially if someone wants to take a look at a specific experiment). Thus, I do not insist on this recommendation and I leave it to the authors’ choice.

Reviewer 2 Report

In the present study, D´Aloia et al seek to characterise and investigate the potential/the overall effect of cannabidiol (CBD) on a human triple-negative breast cancer cell line (namely MDA-MD-231 cells). For this purpose, a great number of different in vivo studies and analyses were performed. A special focus was put on the potential influence of media serum concentrations on CBD effects and delineating the serum components that may be involved. The study is well structured and follows a logical order. The data is interpreted objectively without making „unduly assumptions“. The graphs are well organized and easily understandable. The materials and methods section contains all necessary detail to enable replication of the experiments conducted. The figure legends give excessive detail on how the data was generated exactly.  

-         -  is there any knowledge on the effect of CBD on more complex, 3D, organoid structures of breast cancer cells? Do you assume the effects that you have seen in your one-dimensional cultures would be very much replicable in cultures containing more complexity, hence, getting closer to the actual in vivo situation?

-        -   how do you judge the concentration of CBD needed in your cultures to exert an anti-proliferative effect with regards to potential concentrations that may be needed under in vivo circumstances? Do you think much higher CBD levels will be needed under in vivo circumstances? Do you think major negative effects may come into play when higher concentrations are needed to induce similar effects?
